# Diagnostic and Prognostic Value of PACAP in Multiple Myeloma

**DOI:** 10.3390/ijms241310801

**Published:** 2023-06-28

**Authors:** Tünde Tóth, Hussain Alizadeh, Beáta Polgár, Renáta Csalódi, Dóra Reglődi, Andrea Tamás

**Affiliations:** 1Department of Anatomy, ELKH-PTE PACAP Research Team, Centre for Neuroscience, Medical School, University of Pécs, 7624 Pécs, Hungary; toth.tundi206@gmail.com (T.T.); dora.reglodi@aok.pte.hu (D.R.); 21st Department of Medicine, Division of Hematology, Medical School, University of Pécs, 7624 Pécs, Hungary; alizadeh.hussain@pte.hu; 3Department of Medical Microbiology and Immunology, Medical School, University of Pécs, 7624 Pécs, Hungary; polgar.beata@pte.hu; 4Department of Hematology, Balassa János Hospital of Tolna County, 7100 Szekszárd, Hungary; drcsrenata@gmail.com

**Keywords:** PACAP, multiple myeloma, ELISA, diagnostic and prognostic factor

## Abstract

Pituitary adenylate cyclase-activating polypeptide (PACAP) is a multifunctional neuropeptide with well-known anti-inflammatory, antioxidant, antitumor, and immunomodulatory effects. PACAP regulates the production of various proinflammatory factors and may influence the complex cytokine network of the bone marrow microenvironment altered by plasma cells, affecting the progression of multiple myeloma (MM) and the development of end-organ damage. The aim of our study was to investigate the changes in PACAP-38 levels in patients with MM to explore its value as a potential biomarker in this disease. We compared the plasma PACAP-38 levels of MM patients with healthy individuals by ELISA method and examined its relationship with various MM-related clinical and laboratory parameters. Lower PACAP-38 levels were measured in MM patients compared with the healthy controls, however, this difference vanished if the patient achieved any response better than partial response. In addition, lower peptide levels were found in elderly patients. Significantly higher PACAP-38 levels were seen in patients with lower stage, lower plasma cell infiltration in bone marrow, lower markers of tumor burden in serum, lower total urinary and Bence-Jones protein levels, and in patients after lenalidomide therapy. Higher PACAP-38 levels in newly diagnosed MM patients predicted longer survival and a higher probability of complete response to treatment. Our findings confirm the hypothesis that PACAP plays an important role in the pathomechanism of MM. Furthermore, our results suggest that PACAP might be used as a valuable, non-invasive, complementary biomarker in diagnosis, and may be utilized for prognosis prediction and response monitoring.

## 1. Introduction

Multiple myeloma (MM) is a monoclonal proliferation of plasma cells that primarily affects the elderly population and accounts for 1% of all cancers and 10% of all hematologic malignancies. The average age of patients is 66–70 years, which significantly contributes to the fact that as the disease progresses, treatment options (e.g., cellular therapies) are limited due to the patients’ comorbidities and performance status. For this reason, and because of the clonal heterogeneity of the plasma cells, MM is still considered as an incurable disease [1]. The Global Cancer Observatory estimates that there were 176,404 new cases and 117,077 deaths worldwide in 2020, which means that despite improving mortality statistics, the incidence and prevalence of MM is increasing [2]. The disease develops due to the genetic alteration of plasma cells resting in specific niches of the bone marrow. This occurs through the development of monoclonal gammopathy of unknown significance (MGUS) and smoldering myeloma (SMM), leading to an active MM that requires treatment [3]. Given the affected population, aging is also thought to play an important pathologic role in the development of the disease [4]. After induction therapy, autologous hematopoietic stem cell transplantation (ASCT) provides the longest progression-free survival (PFS) in the treatment setting. However, published data regarding the effect of ASCT in prolonging overall survival (OS) are contradictory [5,6,7]. Despite the initial positive results, relapses occur in a significant proportion of patients. It is crucial to intervene in the progression of the disease before serious organ complications occur, because the survival of MM patients is significantly affected by the degree of end-organ damage. The 5-year survival rate is about 58%, and both PFS and OS can be significantly improved if the disease and the transition from MGUS to MM are detected earlier.

Conventional biomarkers (beta-2-microglobulin (B2M), lactate-dehydrogenase (LDH), M-protein, albumin, cytogenetic alterations) have been included in staging systems. The diagnostic criteria and staging systems proposed by the International Myeloma Working Group (IMWG) remain the key from a disease perspective. However, M-protein is undetectable by serum electrophoresis in 18% of MM cases and by any other diagnostic tool in nearly 3% of patients [8,9]. B2M levels can also be influenced by several other factors (e.g., renal and liver diseases). These data can lead to misdiagnosis and inaccurate staging. Responses in non-secretory myeloma cannot be assessed and monitored by serum and urine tests, even with most sensitive serum free light chain (sFLC) assay. Moreover, most of the conventional prognostic biomarkers used thus far are highly correlated with disease burden. Investigation of the interactions between the bone marrow microenvironment, consisting of both cellular and non-cellular components, is also crucial for predicting prognosis and drug resistance. Thus, it will become increasingly important in the future to identify new, reliable biomarkers of malignancy that can be easily and rapidly used in the clinical setting to diagnose disease, determine prognosis, and monitor response to therapy [10].

Pituitary adenylate cyclase-activating polypeptide (PACAP) has well-known immunomodulatory, anti-inflammatory, and antioxidant effects, therefore, it could be one of the most promising markers in the future [11,12]. The peptide was discovered in 1989 [13] and belongs to the vasoactive intestinal peptide (VIP)/secretin/glucagon peptide family [14]. It exists in two biologically active forms, PACAP-27 and -38, of which PACAP-38 is the dominant form [15]. Its receptors are the specific PAC1 receptor, with eight known splice variants, and VPAC1 and VPAC2 receptors shared with VIP [14,16]. The concentration of PACAP in the plasma is relatively stable in healthy individuals, there are no significant individual differences. Previous studies have shown that its concentration does not depend on gender and is not affected by the female menstrual cycle in the young population between 20 and 40 years [17,18,19].

In recent years, significant alterations of this peptide have been detected in human tissues under various physiological and pathological conditions (e.g., neurological disorders: Parkinson’s disease, posttraumatic stress disorder; cardiac disorders: cardiomyopathies, acute myocardial infarction; kidney disorders: nephrotic syndrome, nephrectomy), in addition to various malignant disorders. These results suggest a potential role of PACAP in the diagnosis, prognosis, and clinical therapy of certain diseases [11,20,21,22].

The beneficial effects of PACAP on renal function in MM patients and its successful use as an antitumor agent have been demonstrated in several in vivo and in vitro studies. It has been described to be protective in proximal tubule cells and to affect signaling pathways involved in osteolysis or osteolytic processes [23,24,25]. Similarly to dexamethasone, PACAP also inhibits the growth of plasma cells [26]. The peptide is involved in the regulation of the production of several proinflammatory mediators (e.g., TNF-α, IL-6, MIP-1α) and may affect the complex cytokine network of the bone marrow microenvironment, which is altered by the MM cells, influencing the course and progression of the disease and the development of various end-organ damage [23,27,28]. The expression of PAC1 receptor mRNA has been detected in human bone marrow stromal cells, MM cells, and proximal tubule cells [23,29]. In a clinical trial, Li and colleagues administered PACAP-38 as a continuous infusion to an 81-year-old patient with active MM. The results were encouraging, as the patient’s free lambda light chain excretion was reduced after starting therapy [19]. The literature suggests that antiproliferative effects on plasma cells (subsequent reduction in light chain production) and both direct (via PAC1 receptor) and indirect (cytokine-mediated antioxidant function) effects on tubule cells may play an important role in mitigating MM-related renal dysfunction.

It is also known that PACAP may play an important role in hematopoiesis and in the development of cells from the mesodermal maturation lineage. The peptide promotes the hematopoietic stem cell population via the PAC1 receptor by increasing the number of cells in the S phase of the cell cycle by enhancing the mRNA expression of cyclin D1 and Ki67 proteins [29]. Furthermore, PAC1 receptor expression is higher on CD34^+^ stem cells and decreases or disappears with cell maturation [29]. It is suggested that PACAP detectable in the bone marrow is of neuronal origin, and sympathetic innervation may be responsible for PACAP-regulated hematopoiesis in the bone marrow [29].

These results suggest an antitumor and renoprotective effect of PACAP in MM and open the possibility of using this peptide in clinical practice. Our aim was to investigate the changes in the PACAP-38 levels in patients with MM to explore the role of this peptide as a potential biomarker in this disease. We correlated the detectable changes in the PACAP-38 levels with other markers used in clinical practice to gain an understanding of the pathogenic role of this peptide.

## 2. Results

### 2.1. Plasma PACAP-38 Levels in MM Patients and Healthy Controls

We found significantly lower endogenous plasma PACAP-38 levels in MM patients (*n* = 66, mean: 208.4 +/− 103.8 pg/mL) compared to healthy controls (C) (*n* = 10, mean: 311.7 +/− 82.19 pg/mL) (*p* = 0.0012) (Figure 1).

Out of our 66 patients, eight were newly diagnosed MM (NDMM) untreated patients who died by the end of our investigation, allowing us to monitor the diagnostic and prognostic value of the PACAP-38 levels. There was also a significant difference in the PACAP-38 values when only these NDMM patients (*n* = 8, mean: 217.4 +/− 84.89 pg/mL) were compared with the control group (*p* = 0.0314).

### 2.2. Endogenous PACAP-38 Levels in Relation to Demographic and Clinical Parameters

No significant difference was found between the patient and control groups regarding either gender or age (*p* > 0.05). The mean age of the patients was 63.97 +/− 9.807 years and mean age of the control group was 62.10 +/− 9.643 years. In our study, we performed correlation analyses between the plasma PACAP-38 values and the ages of the controls and patients. We found a significant weak negative correlation between the patients’ age and PACAP-38 levels (*n* = 66, *p* = 0.0379, *r* = −0.2561, Spearman) (Figure 2) and there was no significant correlation with the controls’ age (*n* = 10, *p* = 0.8567, *r* = −0.06577, Pearson).

Furthermore, we examined the PACAP-38 levels in relation to comorbidities. No significant differences were found between the hypertensive (*n* = 28) and non-hypertensive groups (*n* = 35) (*p* = 0.3459) and the diabetic (*n* = 11) and non-diabetic groups (*n* = 52) (*p* = 0.0743) (Table 1). We also determined the performance status of our patients according to the Eastern Cooperative Oncology Group (ECOG) system and formed four groups: 0 (*n* = 16), 1 (*n* = 26), 2 (*n* = 14), 3 (*n* = 1). No significant difference was found between the tested groups (*p* > 0.05) (Table 1).

### 2.3. Endogenous PACAP-38 Levels in Relation to Current Disease Status

#### 2.3.1. Plasma PACAP-38 Levels in Patients with Active Disease and in Remission

Patients with MM were divided into two groups according to their response to therapy. The active disease cohort (AD) included all treated individuals with refractory or progressive disease, and patients who relapsed after a previous remission (*n* = 17, mean: 137.9 +/− 30.13 pg/mL). NDMM patients were not included in this analysis. Patients in remission (R) included individuals who achieved a partial response (PR), very good partial response (VGPR), complete remission (CR), or minimal residual disease negativity (MRD neg.) during treatment (*n* = 41, mean: 216.6 +/− 108.3 pg/mL). Our study showed that the active disease cohort had significantly lower PACAP-38 levels compared to the R cohort (*p* = 0.0137). In addition, the plasma PACAP-38 levels of patients in groups AD and R were significantly lower compared to the control group (*p* (C vs. AD) < 0.0001, *p* (C vs. R) = 0.0170) (Figure 3).

#### 2.3.2. Endogenous PACAP-38 Levels in Relation to the Depth of Therapeutic Response

We also performed a detailed analysis regarding the depth of therapeutic response. The stable or progressive disease (SD, PD) group included the same patients in the AD group from the previous analysis; patients in remission were further divided into four groups based on their response to treatment as indicated: PR (partial response; *n* = 10, mean: 159.4 +/− 77.42 pg/mL), VGPR (very good partial response; *n* = 13, mean: 194.9 +/− 65.37 pg/mL), CR (complete remission; *n* = 11, mean: 231.8 +/− 111.0 pg/mL), and MRD neg. (minimal residual disease negativity; *n* = 7, mean: 314.6 +/− 148.6 pg/mL). Significant differences were also observed in this analysis. We found that as the depth of response increased, the patients’ PACAP-38 levels steadily increased. The significance shown in the previous figure between the AD and R groups persisted when the SD, PD and MRD neg. groups were examined (*p* = 0.0048) with a similar, but not significant tendency between the SD, PD and CR groups (*p* = 0.0815). In addition, the MRD neg. patients presented significantly higher PACAP-38 levels than the PR patients (*p* = 0.0350) (Figure 4). The significant difference previously found between the MM patients and controls (Figure 1 and Figure 3) gradually disappeared as the PACAP-38 values increased with the depth of therapeutic response (Figure 4). Thus, we did not detect significant differences in the PACAP-38 levels between patients with VGPR (*p* = 0.2083), CR, and the healthy controls (*p* > 0.9999) and between the MRD neg. group and healthy controls (*p* > 0.9999). The PACAP-38 concentrations in the MRD neg. patients were almost identical to the PACAP-38 concentrations measured in the healthy control group (control mean: 311.7 +/− 82.19 pg/mL) (Figure 4).

We also examined the extent to which the baseline PACAP-38 levels in the NDMM patients predicted whether the patient would achieve CR. We divided these patients into two groups: at least CR (*n* = 3, mean: 316.4 +/− 20.18 pg/mL) and less than CR (*n* = 5, mean: 158.0 +/− 25.59 pg/mL). Despite the small number of elements, our preliminary results showed higher baseline PACAP-38 values in patients who reached CR. However, we found a significant difference between patients with refractory (*n* = 4, mean: 153.2 +/− 26.81 pg/mL) and non-refractory MM (*n* = 4, mean: 281.6 +/− 71.47 pg/mL) (*p* = 0.0302).

### 2.4. Changes in Endogenous PACAP-38 Levels in Relation to Plasma Cell Infiltration in Bone Marrow

Our next goal was to examine the alteration of the plasma PACAP-38 levels in relation to plasma cell infiltration in the bone marrow, which was assessed by histological analysis (Figure 5a) and flow cytometry (FCM) (Figure 5b) to better estimate the proportion of clonal plasma cells. In both analyses, we found a significant moderate negative correlation between the percentage of plasma cells in the bone marrow and plasma PACAP-38 levels (histological results: *n* = 41, *p* = 0.0020, *r* = −0.4681; FCM: *n* = 32, *p* = 0.0030, *r* = −0.5078).

### 2.5. Examination of Endogenous PACAP-38 Levels in Relation to End-Organ Damage

#### 2.5.1. Plasma PACAP-38 Levels in Relation to Bone Lesion and Serum Calcium Levels

No significant relationship was found between either the serum calcium and PACAP-38 levels (*n* = 43, *p* = 0.3657, *r* = 0.1414, Spearman), or between patients with (*n* = 35) and without (*n* = 26) bone lesions. However, the low *p*-value may indicate a trend toward lower plasma PACAP-38 levels in patients with bone lesions (*p* = 0.0626) (Table 2).

#### 2.5.2. Plasma PACAP-38 Levels in Relation to Kidney Disease

No significant association was found between the plasma PACAP-38 levels and MM-related renal disease either in the study of laboratory parameters (carbamide (*n* = 65, *p* = 0.4363, *r* = −0.09823, Spearman), creatinine (*n* = 65, *p* = 0.9126, *r* = 0.01388, Spearman), glomerular filtration rate (GFR) (*n* = 65, *p* = 0.9381, *r* = 0.009821, Spearman) indicative of end-organ damage, or when the patients were divided into groups with (*n* = 43) and without renal impairment (*n* = 22) (*p* = 0.6448) (Table 2). We compared patients with (*n* = 6) and without (*n* = 59) acute renal failure, but no significant difference was found (*p* = 0.3108) (Table 2). We also formed six groups according to the stage of chronic kidney disease (CKD): normal (*n* = 22), CKD1 (*n* = 4), CKD2 (*n* = 13), CKD3 (*n* = 19), CKD4 (*n* = 3), and CKD5 (*n* = 4), but we found no significant differences between the tested groups (*p* > 0.9999) (Table 2).

#### 2.5.3. Plasma PACAP-38 Levels in Relation to MM-Associated Anemia

In the case of anemia associated with MM, we examined the connection between the plasma PACAP-38 levels and laboratory values indicative of anemia [hemoglobin (Hgb), hematocrit (Htc)]. We formed two groups based on whether the patient was anemic or not, according to the definition of MM-related anemia (Hgb below 100 g/L). In the present study, we found no significant correlation or association in either case [Hgb (*n* = 64, *p* = 0.8737, *r* = −0.02027, Spearman), Htc (*n* = 64, *p* = 0.7210, *r* = −0.04552, Spearman); between anemic (*n* = 12) and non-anemic (*n* = 52) groups (*p* = 0.6898)] (Table 2).

#### 2.5.4. Plasma PACAP-38 Levels in Relation to Extramedullary Disease, Plasma Cell Leukemia and Other End-Organ Damage

We found no significant difference between patients with extramedullary disease (EMD) or plasma cell leukemia (PCL) (*n* = 18) as a result of the evolutionary process of MM disease and patients who were unaffected (*n* = 48) in this regard (*p* = 0.3488). In addition, there was no significant difference between patients with MM-related amyloid light-chain (AL) amyloidosis (renal or cardiac involvement) (*n* = 4) and non-affected individuals (*n* = 62) (*p* = 0.1728) (Table 2).

### 2.6. Plasma PACAP-38 Levels in Relation to Staging and Risk Classifications Used for MM

Based on the recommendations of the IMWG for the staging and risk classification of MM, endogenous PACAP-38 levels were examined according to the International Staging System (ISS), Revised International Staging System (R-ISS), and cytogenetic risk classification [30,31,32]. For the analysis, only the PACAP-38 levels of those patients who were diagnosed with MM disease within one year of sample collection were examined due to the low number of NDMM patients. For the R-ISS and cytogenetic risk scores, we found no significant difference between the tested groups (*p* > 0.05), but there was a decreasing trend between Stages I and II (*p* = 0.1547) (Table 3). For the ISS, we formed three groups in relation to the stages of MM. As shown in the figure, the PACAP-38 levels decreased progressively in the higher stages (Stage I: *n* = 4, mean: 305.8 +/− 147.1 pg/mL, Stage II: *n* = 13, mean: 175.9 +/− 72.33 pg/mL, Stage III: *n* = 14, mean: 143.9 +/− 38.59 pg/mL). Significant differences were found between the PACAP-38 levels of Stage I and Stage II (*p* = 0.0108) and between Stage I and Stage III patients (*p* = 0.0014) (Figure 6).

### 2.7. Endogenous PACAP-38 Values in Relation to Laboratory Parameters

In our study, we also investigated the relationship between various laboratory parameters important in MM and plasma PACAP-38 levels. Correlation tests showed a significant, strong negative correlation between B2M (above 5.5 mg/L, *n* = 9, *p* = 0.0255, *r* = −0.7500), and moderate negative correlation between Bence-Jones (BJ) protein (*n* = 19, *p* = 0.0180, *r* = −0.5359) and PACAP-38 levels. Besides, we found a negative trend for PACAP-38 and erythrocyte sedimentation rate (ESR) (*n* = 25, *p* = 0.0833, *r* = −0.3531), urinary total protein (UTP) (*n* = 23, *p* = 0.0522, *r* = −0.4191) and serum kappa/lambda ratio (KLR) above 1.65 (*n* = 18, *p* = 0.1144, *r* = −0.3849), but there was a positive trend with KLR below 0.26 (*n* = 8, *p* = 0.1323, *r* = 0.5952). In correlation tests, no significant relation was found between serum LDH, M-protein, albumin, total protein (STP) and PACAP-38 levels (Table 4). However, when two groups were formed according to the LDH serum concentrations (225 U/L was chosen as the cutoff between abnormal (*n* = 43) and normal (*n* = 7) LDH values), a substantial decrease in plasma PACAP-38 was observed in patients with higher LDH values (*p* = 0.0339) (Table 5). A significant difference was also found when the UTP was considered and two groups (above (*n* = 15) and below (*n* = 8) 0.2 g/L protein excretion) were compared (*p* = 0.0282) (Table 5), confirming the trend of the previous correlation analysis, which revealed lower PACAP-38 values in patients with higher UTP (Table 4).

### 2.8. Plasma PACAP-38 Levels in Relation to Therapy of MM

#### 2.8.1. Plasma PACAP-38 Levels in Relation to Treatment

We also analyzed the PACAP-38 values of our patients in relation to combination treatment and we formed three groups: VTD (bortezomib–thalidomide–dexamethasone) (*n* = 18, mean: 172.9 +/− 72.09 pg/mL), RVd (lenalidomide–bortezomib–dexamethasone) (*n* = 11, mean: 258.1 +/− 107.4 pg/mL), and VCD therapies (bortezomib–cyclophosphamide–dexamethasone) (*n* = 6, mean: 146.3 +/− 44.34 pg/mL). The PACAP-38 values of patients treated with RVd were much higher than those of the other groups (*p* = 0.0307) (Figure 7).

The PACAP-38 values of our treated patients were also analyzed with respect to the therapeutic agents associated with the treatments, and the obtained results are shown in the Table 6. We found a significant difference between the lenalidomide-treated (*n* = 18) and the untreated group (*n* = 40), with the treated cohort presenting higher PACAP-38 values (*p* = 0.0062). However, the alkylating agents-treated group (*n* = 10) presented lower PACAP-38 values than the untreated group (*n* = 48) (*p* = 0.0200). The PACAP-38 values of patients treated with daratumumab (*n* = 4) were also higher, but probably due to the small number of elements, no significant difference could be detected (*p* = 0.0773) (Table 6).

#### 2.8.2. Plasma PACAP-38 Levels in Relation to Stem Cell Mobilization

Several plasma samples were collected from the patients admitted to the clinic for stem cell mobilization and subsequent peripheral stem cell collection (PBSC). Samples were collected first before the start of conditioning treatment, and second, during stem cell collections. A total of nine patients with MM were observed throughout the study; in six of these patients, the collection had to be performed in two sessions (PBSC1 and PBSC2), and in another two patients, in three sessions (PBSC1, PBSC2, and PBSC3). Therefore, samples collected at different times were as follows: before conditioning (BC; *n* = 9, mean: 150.1 +/− 91.61 pg/mL), at first peripheral stem cell collection (PBSC1; *n* = 9, mean: 275.3 +/− 175.6 pg/mL), and during peripheral stem cell collections thereafter (PBSC2; *n* = 8, mean: 329.6 +/− 166.2 pg/mL; PBSC3; *n* = 2, mean: 694.1 +/− 177.3 pg/mL). We observed that the endogenous PACAP-38 levels were significantly elevated throughout the process compared to the baseline (BC: mean: 150.1 +/− 91.61 pg/mL) (*p* = 0.0459) (Figure 8).

### 2.9. Plasma PACAP-38 Levels in Relation to Survival of Patients

When we examined the survival of NDMM patients, we found a strong positive correlation between the plasma PACAP-38 levels and OS (time from diagnosis to death in months; *n* = 8, *p* = 0.0131, *r* = 0.8383, Spearman) (Figure 9a), and PFS (time in months from the start of induction treatment to progression or death from any cause; *n* = 8, *p* = 0.0190, *r* = 0.8301, Spearman). All NDMM were treatment-naive at the time of sampling. After that, the majority of our NDMM patients (*n* = 6) were subsequently treated with VTD therapy, had high ISS and R-ISS stage (III), and bone marrow plasma cell infiltration was greater than 50%. More than half of our patients (*n* = 5) had a high cytogenetic risk.

We also found a significant moderate positive correlation between PACAP-38 and PFS in patients who had undergone ASCT (*n* = 13, *p* = 0.0203, *r* = 0.6445, Spearman) (Figure 9b). PFS was calculated as the time from ASCT to progression or death. All patients received a single line of induction therapy before transplantation and received a single ASCT. All patients received lenalidomide monotherapy as a maintenance after ASCT, according to the IMWG guidelines [33].

The PACAP levels also showed a significant moderate positive correlation with PFS (time from sample collection to disease progression or death) in CR (*n* = 11) and MRD neg. patients (*n* = 7). We did not have detailed and accurate information on the outcome of one MRD neg. patient, therefore data from this patient were excluded from this analysis (*n* = 17, *p* = 0.0377, *r* = 0.5114, Spearman) (Figure 9c).

### 2.10. Investigation of the Specificity and Sensitivity of Plasma PACAP-38 Level as a Biomarker in MM

When the receiver operating characteristic (ROC) curve was plotted for the active disease patients (NDMM + AD patients, *n* = 25) and controls (*n* = 10), the area under the ROC curve (AUC) was 0.936 (95% confidence interval (CI): 0.799-0.991), *p* < 0.0001 (Figure 10), indicating an outstanding diagnostic performance of the test. Based on the highest Youden index, the calculated cut-off value was ≤186.729 pg/mL with a specificity of 100% and a sensitivity of 88%.

## 3. Discussion

The purpose of this study was to detect alterations in the PACAP-38 levels in patients with MM to investigate the role of this peptide as a potential biomarker. We correlated the PACAP-38 levels with other markers used in clinical practice to gain a more detailed understanding of the pathomechanisms underlying the abnormalities seen in this disease.

For a long time, the presence of disease-related end-organ damage was a diagnostic cornerstone of the definition of symptomatic MM [34]. However, in 2014, the criteria were modified by the IMWG to include three new biomarkers (more than 60% plasma cells in bone marrow, or more than 100 free light chain ratio, or more than one focal lesion on MRI scans) [35,36]. The latter modification was necessary because a randomized trial showed that the treatment of patients with high-risk SMM could improve their survival [37]. This fact represents a paradigm shift in the approach of the disease, as these changes allow us to intervene in the development and progression of MM before end-organ damage sets in. Biomarkers such as LDH or B2M correlate with tumor burden and have a prognostic role. Furthermore, these conventional biomarkers have several limitations as prognostic markers. At the same time, the incidence and prevalence of MM are steadily increasing, and survival has not improved significantly in recent years. Presumably, survival can be further improved by the earlier detection of transformation from pre-symptomatic stages to symptomatic disease using additional sensitive markers [8,9,10]. Recently, a number of new biomarkers have been discovered, whose use is becoming more widespread, and allow us to identify the genetic abnormalities underlying MM disease [38]. However, tumorgenesis in MM is highly dependent on the local tissue microenvironment and its alterations. The expression of extracellular matrix proteins (e.g., ANXA2, LGALS1, LAMB1, ITAG9) is altered at the gene and protein levels in MGUS, SMM, and MM. Therefore, their investigation may be critical for both the prognosis and detection of subsequent drug resistance [10,39,40,41]. Some studies have reported that despite successful antitumor induction, the inflammatory environment formed by tumor cells persists and provides a perfect basis for the progression of residual disease. The best therapeutic response can be expected from a therapeutic approach that simultaneously targets tumor plasma cells and the complex bone marrow microenvironment [42]. With the development of proteomic techniques, it is possible to screen for several proteins in MM that might reveal the altered expression of several factors [43] that are necessary for the prognostic evaluation of MM for the use of personalized biology-based treatments and for the appropriate monitoring of therapeutic response [44]. These next-generation biomarkers (e.g., angiogenesis markers, miRNA, proteomic markers, immunological markers) need further validation to be incorporated into clinical practice [10].

Our research suggests that PACAP may be one of these promising markers in the future, which may complement conventional diagnostic and prognostic biomarkers. PACAP is a multifunctional neuropeptide with proven anti-inflammatory, antioxidant, and immunomodulatory effects. The level of endogenous PACAP varies considerably in different diseases, so changes in PACAP levels serve as diagnostic and prognostic markers and may also be helpful in planning clinical therapies for certain diseases [11]. In malignancies, a decrease in the tissue PACAP level was described in colon carcinoma [45], non-small cell lung cancer [45], renal tumors [46], papillary thyroid carcinoma [47], pituitary adenoma [48], pancreatic ductal adenocarcinoma [49], and insulinoma [50], while an increase was observed in neuroblastoma [51,52], ductal and lobular breast carcinoma [53], cervical cancer [54], and prostate tumor [46]. In contrast, no significant difference was found in bladder and testicular tumors compared to healthy tissue [46]. The antitumor activity of the peptide has been described in MM, and PACAP has also been shown to be protective in the development of osteolytic bone destruction [26] and disease-related renal injury [23,24]. PACAP has also been described to affect signaling pathways directly involved in MM cell survival and disease progression [24]. In addition, expression of its receptors on bone marrow stromal cells and on proximal tubule cells as well as on MM cells has been described, showing that PACAP also affects the homeostasis of the bone marrow microenvironment [24,29]. PACAP has already been shown to contribute to hematopoiesis, affecting various cell cycle regulatory factors via the PAC1 receptor [29].

In our study, we demonstrated a significant decrease in the PACAP levels in patients with MM compared with age- and gender-matched healthy controls, which may open up the future possibility of using PACAP as a potential biomarker for the diagnosis of MM.

We found a significant negative correlation between the patients’ age and PACAP levels. The results of several in vivo experiments suggest that the dysregulation of PACAP plays an important role in aging, and the level of this neuropeptide progressively decreases with age. PACAP knockout mice show signs of premature aging, even at a young age. Age-related degenerations due to increased neuronal vulnerability, apoptosis, oxidative stress, and inflammatory processes have been observed much earlier in these animals than in their wild-type counterparts [55]. PACAP deficiency leads to earlier retinal lesions, corneal keratinization, and blurring [56]; senile systemic amyloidosis [57]; early cartilage degeneration [58]. Although the significant correlation between PACAP levels and age has not been described in either healthy or patient populations, recently, our research group demonstrated significantly lower PACAP levels in a population of Parkinson’s disease patients older than 50 years. In this case, however, an increase in younger patients due to deep brain stimulation could not be excluded [17,20]. Thus, our study is currently the first in clinical research to demonstrate a significant decrease in the plasma PACAP levels in correlation with the age of the MM patients. While the probability of developing cancer at a young age is 1 in 29, this ratio increases to 1 in 3 at age 70 [59]. Therefore, aging is an important risk factor for most cancers, especially MM [60], which has been confirmed by the fact that the incidence of MGUS increases steadily with age [61]. The main risk factors for the development of this disease are age-related immunological changes at the level of terminally differentiated plasma cells, which may lead to stochastic, genetic, epigenetic, and cellular processes that ultimately result in the clonal selection and expansion of MM cells. In these processes, PACAP may play a key role by reducing oxidative stress, preventing DNA damage, and the development of structural and numerical chromosomal aberrations characteristic of the disease. We hypothesize that the lower PACAP levels result in reduced tumor control as the anti-inflammatory, antioxidant, and immunomodulatory properties of PACAP may play a protective role against tumor proliferation. Aging processes and many harmful environmental variables may presumably cause a pathological reduction in PACAP with age [62,63]. However, the evolution of manifest MM from asymptomatic forms (MGUS, SMM) is an extremely slow process, and prolonged progression leads to clonal heterogeneity of tumor cells, which also significantly compromises the efficacy of therapy and limits clinical options [4]. Therefore, treating elderly, polymorbid patients with poor performance status is a major challenge [64,65].

No significant association was found between hypertension, diabetes, ECOG performance status, and PACAP levels, suggesting that the most common comorbidities of our patients and their current performance did not significantly affect the results of our study.

In our cross-sectional study, some patients had active disease and others were in remission, therefore, we could assess their current disease status. It was found that patients with active disease had significantly lower PACAP levels. Further grouping of patients in remission according to the depth of response to therapy showed that the PACAP values increased steadily with the depth of response until the PACAP values of MRD neg. patients (which is the best prognostic factor [66,67]) reached the PACAP values of the healthy controls. Among the NDMM patients, those with lower PACAP values reached CR less frequently, and there were also more refractory patients. These results confirm our hypothesis that PACAP may be a promising biomarker for complementing information from traditional biochemical and imaging criteria of response.

The significant negative correlation we found between PACAP and the percentage of clonal plasma cells may also be helpful in assessing therapeutic response. This result is consistent with previous studies by Li and Arimura. It has been nearly 20 years since they described that PACAP has antineoplastic effects and directly inhibits plasma cell proliferation. The anti-apoptotic effect of PACAP is well-known, but several studies have already described that PACAP inhibits the proliferation and growth of malignant cells in certain tumors (e.g., neuroblastoma) [68]. The expression of peptide receptors has already been described on different types of tumor cells. However, many variants of the PAC1 receptor are known through alternative splicing. PACAP inhibits cell proliferation through the cAMP-dependent pathway by the short variant of the PAC1 receptor, while the growth, reproduction, and survival-stimulating effect of the peptide is mediated through the splice variant (SV) 2 subtype of the human PAC1 receptor activating PLC-dependent mechanisms and the MAPK pathway. The PAC1 receptor short, SV1, and SV2 variants were discovered on bone marrow stromal cells. On MM cells, only the short PAC1 receptor variant has been described among these receptors [26,69]. Apoptosis and cell cycle dysregulation are also pathogenic factors in MM, which could be affected by PACAP through the regulation of cyclin D1, Ki67, and various anti-apoptotic proteins (e.g., Bcl-2) [29]. Previous research and the results of our study suggest that PACAP restores cell cycle control in MM and like dexamethasone, can induce apoptosis in MM cells. It has also been described that PACAP sensitizes MM cells to DNA-damaging therapeutic agents, and when used in combination with dexamethasone, it has a synergistic effect by enhancing caspase-9 activation via cytochrome c. PACAP could also affect the activation of MYC via the PI3K-mTOR pathway and the function of DNA repair systems by affecting cyclin D [19,23,24,25,26,29]. Moreover, the peptide decreases the levels of IL-6 and TNF-α, leading to the inhibition of the p38-MAPK and NF-κB signaling pathways, which play important roles in plasma cell proliferation, migration, and survival [19,23,24,25,26]. The prognostic value of PACAP is also supported by the negative correlation with plasma cell bone marrow infiltration, as it is known that a high proportion of plasma cells is associated with shorter OS [70,71]. In addition, the literature data suggest that patients with a plasma cell count below 5% before transplantation can expect a better therapeutic response and longer survival after ASCT [72].

No significant difference was found for the end-organ damage test. Thus, it is unlikely that the presence of organ damage had a significant effect on our studies. However, during the investigation of bone lesion, the low *p* value already indicated a trend for a difference between the two groups of studied patients, suggesting that PACAP levels may be lower in patients with bone lesions. This result may be explained by the association of PACAP and MIP-1α, which plays an important role in the development of osteolytic bone destruction by osteoclast activation and may affect patient survival [73,74,75,76,77]. It has been described that PACAP can inhibit the expression of MIP-1α already at the mRNA level, suggesting that PACAP may play an important protective role in the development of MM-induced osteolysis [27,78,79].

We also tested PACAP values for the IMWG-recommended risk and staging classification systems used in our patients, and only in ISS have we found significant differences. A significant decrease in PACAP levels was observed in higher ISS stages, which may indicate a potentially valuable prognostic biomarker role of PACAP. For R-ISS, there was already a trend toward a decline between Stages I and II, but no significant difference was observed between the two groups. This result is probably due to the fact that ISS uses B2M and albumin to assess stage, whose values are significantly related to the disease burden [30,31]. At the same time, we must mention that we could not properly evaluate the examination of these stages and risk classifications due to the small number of NDMM patients (*n* = 8). Changes in the PACAP levels due to the treatment of patients might have biased these results.

For B2M, a parameter for assessing staging according to ISS, a significant negative correlation with PACAP was described at elevated levels, with a very strong *r* value. In the case of LDH, an indicator of proliferative activity as well as the tumor mass itself, we have shown that lower PACAP values are associated with higher LDH values. Elevated values of both laboratory parameters serve as poor prognostic markers [80,81,82]. When we examined various serum and urine proteins that indicate the disease progression of MM, we found significant changes between the BJ protein, UTP, and PACAP. Increased levels of both protein levels were observed in correlation with lower PACAP levels. Our results on the BJ protein confirm previous data in the literature, as Li et al. reported in their case study that continuous PACAP infusion reduced the patient’s free lambda light chain excretion [19]. The amount of BJ protein refers to two things: one is the extent of renal damage, and the other is the tumor mass [83,84,85,86,87]. When these proteins were examined, patients showed renal damage, irrespective of UTP. Renal function fell within the normal GFR range in only one patient. Considering this information, we suggest that this result is not related to the renal function of the patients, but to the amount of paraproteins produced by the MM cells. The most accurate method currently available to detect paraproteins in serum is the sFLC, in which the KLR was measured. A negative trend between PACAP and KLR above 1.65 and a positive trend below 0.26 was observed. This result was confirmed by the correlation with the amount of light chains in urine. In our study, there was also a negative trend between PACAP and ESR. The altered albumin/globulin ratio indicated by ESR also suggests that PACAP must affect the amount of paraproteins produced by the plasma cells. Furthermore, ESR is an independent marker of MM, whose elevated level is associated with poor prognosis. The opposite trend to PACAP also confirms the potential role of this neuropeptide as a biomarker [88,89].

We examined the relationship between PACAP and therapeutic procedures in MM. In the case of combination treatments, the highest PACAP levels were measured for the RVd protocol. When we analyzed the treatments, we could not exclude the possibility that the response to RVd, which was slightly better, had no effect on the PACAP levels. However, when we also examined the PACAP levels measured in relation to the depth of response to therapy, we found higher peptide levels in the AD, CR, and MRD neg. subgroups as a result of the RVd treatments. There have been several studies comparing the therapeutic protocols, and there is evidence to suggest that RVd may be associated with better outcomes compared to VTD. RVd is associated with a higher overall response rate, a longer time to disease progression, and a longer PFS to VTD. RVd also showed improved efficacy and tolerability compared to VTD [90,91]. It is known that the RVd protocol is the most effective among the currently used antimyeloma therapies. It is listed in the first line in the EHA-ESMO 2021 guideline because this protocol increases OS and PFS the most [5,33]. The RVd protocol differs from the others in that it contains lenalidomide, which also has immunomodulatory effects. When therapeutic agents were examined, lenalidomide was the only agent that showed a significant correlation with an increase in the PACAP levels, while in the case of other agents, the PACAP levels decreased. The effects of immunomodulators (IMiDs) are highly variable: they increase the expression of tumor suppressor genes, arrest the cell cycle, inhibit angiogenesis, reduce the levels of adhesion molecules, inhibit the adhesion of MM cells to bone marrow stromal cells, suppress osteoclast formation and function, and decrease the expression of proinflammatory cytokines. IMiDs also affect the immune system’s ability to recognize tumor cells [5,92,93]. Although thalidomide, lenalidomide, and pomalidomide have a similar biological activity, the latter two are more effective [94]. Bone marrow stromal cells are of particular importance in supporting MM cells. MM cells modulate the bone marrow microenvironment by inducing the production of cytokines that promote plasma cell survival and proliferation, and this inflammatory environment persists after remission [42]. Regarding the study of these cytokines, it is already known that both PACAP and lenalidomide have effects that lower the levels of these factors (IL-6 and TNF-α), thus attenuating the processes in the microenvironment that contribute to the survival of MM cells [23,92,93]. A similarly promising and effective agent is daratumumab, which also showed an increasing trend in PACAP, but presumably due to the small number of elements, no significant difference was detected. The explanation for the decrease in PACAP with other therapeutic agents, especially alkylating agents (cyclophosphamide, melphalan), requires further investigation. There is no extensive literature on this topic yet, but it is known that low-dose PACAP treatment protects against cyclophosphamide-induced thymic atrophy. Cyclophosphamide also increases PAC1 receptor expression. These results suggest that alkylating agents may lead to a decrease in endogenous PACAP levels [95].

For the therapies used in MM, we also examined how the PACAP levels behaved during stem cell mobilization and collection. This showed that the PACAP levels gradually increased during the process, which is likely to be related to G-CSF stimulation [33]. The exact role of PACAP in hematopoiesis is still unclear, but previous research has indicated that PACAP may contribute to the regulation of hematopoiesis in the bone marrow through its receptors and affects the function and maturation of CD34^+^ hematopoietic stem cells [29]. Previous studies have suggested that PACAP negatively regulates megakaryopoiesis in conditions where hematopoiesis is already impaired. PACAP is known to have a thrombopoietic effect by inhibiting the VPAC1 pathway. However, there are no consistent results or studies on erythropoiesis or lymphopoiesis [96,97,98,99].

In the last part of our study, we also found a significant positive correlation between PACAP, OS, and PFS. However, the great number of refractory patients among NDMM patients may have biased this association between PACAP and PFS. Therefore, we collected follow-up data from patients with few months earlier diagnosis who were about to undergo their first ASCT at the time of our sample collection as well as from patients who reached at least CR. In these cases, we also found significant positive associations between PFS and PACAP, supporting the potential prognostic role of peptides in MM [8,100].

We also investigated the diagnostic value of PACAP levels in distinguishing MM patients from healthy individuals. The ROC analysis revealed an excellent diagnostic performance of PACAP in MM with an AUC of 0.936, specificity of 100%, and sensitivity of 88%. The PACAP cut-off value determined in our study was: ≤186.729 pg/mL. Other diagnostic (sFLC) and prognostic (B2M, LDH) biomarkers previously used did not exceed the specificity we demonstrated, but their sensitivity was similar to our results (sFLC: AUC: 0.875, cut-off value of involved/uninvolved light chain ratio: 3.2121, with a specificity: 81.16%, sensitivity: 94.87%; AUC values for B2M, LDH and total amount of immunoglobulins (Ig) were 0.843, 0.547 (nonsignificant), and 0.723; the cut-off values of the three biomarkers were B2M: 1.95 mg/L, LDH: 220 U/L, and Ig: 46.4 g/L). The diagnostic value was better for the combination of three biomarkers (sFLC, B2M, Ig): AUC: 0.952; sensitivity: 94.20%; specificity: 86.75%. The addition of LDH did not optimize the screening value [101]. At the same time, we should not ignore the limitations of our study. Our results need to be confirmed on a larger sample of patients and with validated measurement tools. However, the addition of PACAP to these combinations is expected to further increase the sensitivity and specificity.

Our results indicate that PACAP may play an important role in MM as we demonstrated that PACAP levels were lower in patients compared with the healthy control group and that the PACAP levels gradually decreased with age in patients with MM, which may lead to reduced tumor control. Indeed, the peptide may affect plasma cells directly via the short variant of the PAC1 receptor, and indirectly by affecting factor expression in the bone marrow microenvironment, resulting in decreased proliferative capacity, survival, migration, and adhesion [26,69]. However, we do not yet know what exactly causes the decline in PACAP levels. Presumably, aging processes and various deleterious environmental factors may lead to a pathological decline in PACAP with age [62,63]. It is also possible that the decreased PACAP level in this disease is directly due to the altered tumor cell microenvironment [24,29]. This is also suggested by our finding that PACAP increased with the deepening of therapeutic response, and with a reduction in the disease burden until the mean and median values of MRD neg. reached those of the healthy age- and gender-matched control group. We cannot exclude the possibility that these two factors occur together. We believe that the decrease in PACAP levels with age favors the development of the disease, during which plasma cells have an adverse impact on the microenvironment, causing a subsequent drop in peptide levels and a corresponding decline in tumor control. In addition to these factors, decreased PACAP level also influences the development and progression of disease-related organ damage such as renal failure and bone lesions, as a result of the peptide’s antiproliferative action as well as the protective effect in proximal tubule cells and the inhibitory effect on osteoclasts [23,24,25].

We have shown that PACAP is closely related to disease burden, as our results indicate that patients with active disease, higher plasma cell infiltration in bone marrow, higher tumor markers (LDH, B2M, BJ protein), and higher ISS stage have lower PACAP levels. However, these results do not imply that the peptide directly and exclusively affects the number of plasma cells. This is also supported by the fact that although most of our NDMM patients had a similar disease burden, those with higher PACAP levels achieved a deeper response and prolonged survival. Examining the PFS of our patients who achieved CR or MRD neg., we also found a significant positive correlation with the PACAP levels. We suggest that the plasma PACAP level could reflect not only the disease burden, but also the extent of the inflammatory changes in the bone marrow microenvironment that persist after hematological remission and may promote disease recurrence [42].

The prognostic role of PACAP is clearly evident, and in the future, it may also provide a new dimension to the diagnostic process (e.g., non-secretory MM). Since it is known that the synthesis of various microenvironmental proteins changes in MGUS, SMM, and MM, it is reasonable to assume that PACAP levels may vary over the course of progression from premalignant states to symptomatic disease, and that this peptide may represent an early predictor of disease progression. Unquestionably, this assumption of ours requires further investigation and expansion of the study group to include individuals suffering from these premalignant diseases.

In conclusion, these results confirm our hypothesis that PACAP plays an important role in the pathogenesis of this disease and support the use of this peptide in clinical practice as a valuable biomarker.

## 4. Materials and Methods

### 4.1. Selection of Patients for the Study

The study included patients with confirmed diagnosis of MM in the Division of Hematology, 1st Department of Medicine, Medical School, University of Pécs. Sample collection and patient data follow-up were conducted between July 2019 and January 2023. A total of 66 patients with MM were included in the examination, most of whom were on treatment. At the time of the study, there were a total of eight patients who had not received treatment. The mean age of the patients was 63.97 +/− 9.807 years; 35 women and 31 men participated in the study. The youngest patient was 39 and the oldest was 84 years old. The disease characteristics of our patients and the treatments they received are summarized in the following table (Table 7). The control blood samples were taken from mostly healthy volunteers who did not suffer from hematological diseases, but three of them had hypertension; the others were completely healthy and were not on any medications. The mean age of the control group was 62.10 +/− 9.643 years, the oldest participant was 77 years old, and the youngest was 47 years old. The study involved six women and four men.

### 4.2. Collection and Handling of Blood Samples and Patient Data

Several routine laboratory and histological tests were performed in most of the participants: total and ionized calcium, renal function (serum creatinine and urea nitrogen or carbamide levels), inflammatory parameters (serum C-reactive protein level and ESR), complete blood count (Hgb, Htc, white blood cell count (WBC), red blood cell count, platelet count), LDH, serum and urine protein determinations (electrophoresis, immunofixation and light chain assay: STP and UTP, albumin, B2M, BJ protein, M-protein, KLR), and the percentage of plasma cells in bone marrow (analyzed with histological and FCM examinations) were determined. These markers can be used to monitor the progression of disease and organ damage. Where these tests were not performed on the day of sampling, the results were analyzed retrospectively up to 1 month for laboratory tests and up to 6 months for histological reports. All laboratory tests were performed in the Department of Laboratory Medicine, and Department of Pathology, Medical School, University of Pécs. In addition, patients with MM were evaluated according to age, gender, OS and PFS, disease status, ECOG score, different staging (ISS, ISS-R) and risk classifications (cytogenetic risk), presence of end-organ damage or myeloma defining event: anemia, hypercalcemia, bone lesion, renal failure. Furthermore, we considered the therapeutic protocol according to which the patients were treated, whether they underwent stem cell transplantation or not, and if the patient was submitted for mobilization and collection of stem cells, this process was also accompanied by sampling.

For PACAP-38 determination, peripheral venous blood was taken in tubes including EDTA (ethylene-diamine tetra-acetic acid). Because of the presence of dipeptidyl-peptidase IV (DPPIV) in the plasma, a protease inhibitor (20 μL of aprotinin solution at 1.4 mg/mL for 1 mL of blood) was added to the blood samples and an ice water bath was used for storing the tubes to avoid peptide degradation. The EDTA tubes were centrifuged twice immediately after the collection (first 1000 rpm, 4 °C, 5 min; then 3500 rpm, 4 °C, 15 min), then the supernatant was collected and stored in polypropylene tubes (Sarstedt, Budapest, Hungary) at −80 °C until ELISA analysis.

Patients with MM had their blood drawn once (6 mL blood in an EDTA tube) and their disease stage and treatments were accurately documented. Patients (*n* = 9) who came for stem cell mobilization had a blood sample drawn on the first day of their hospital admission (6 mL blood in an EDTA tube). Thereafter, most patients received chemotherapy according to the mobilization protocol [intermediate dose of cyclophosphamide and following that G-CSF injection (1 × 48 million E/0.5 mL solution sc. injection daily for 8–11 days)]. The patients’ WBC count was continuously monitored, and after the aplasia period, if the absolute neutrophil count exceeded 2 G/L, blood was drawn and the CD34^+^ cell count was determined by FCM. When the CD34^+^ cell count in peripheral blood reached ≥20/μL, the patient underwent cytapheresis. At the same time, an additional 6 mL of blood was collected in an EDTA tube to measure the endogenous PACAP-38 levels by ELISA. If one collection did not yield at least one graft sufficient for one autologous stem cell transplantation—two in younger patients—the procedure was repeated the next day. If the collections were repeated, we also repeated our sampling.

If the MM patient was submitted for the mobilization and collection of stem cells, only those samples were included in the baseline MM samples, which were collected before the start of the mobilization protocol because a preliminary study by our research group showed that PACAP-38 was increased in plasma samples during the process. Therefore, samples from patients who had already received mobilization chemotherapy and G-CSF injections at that time were excluded from this part of the study. These samples were evaluated in a follow-up study as a series of contiguous to evaluate changes in the PACAP-38 levels during mobilization and collection in individual patients.

All human sample collections were carried out according to a protocol approved by the Institutional Ethics Committee of University of Pécs (PTE KK 6383) and followed the Declaration of Helsinki and International Conference on Harmonization (ICH) Good Clinical Practice (GCP) Guidelines to protect the rights of human subjects. In all cases, we obtained informed consent of the volunteers.

### 4.3. Measurement of PACAP-38-like Immunoreactivity (LI) by ELISA

A conventional, sandwich type enzyme-linked immunosorbent assay (ELISA) (Human PACAP-38 ELISA Kit, MyBiosource, San Diego, CA, USA) was performed to quantify the PACAP-38-like immunoreactivity (LI) in the collected clinical samples. According to our earlier studies [21], the determined concentrations are referred to as PACAP-38 levels or values in the manuscript. The assay was performed according to the manufacturer’s instructions. Briefly, before the assay, frozen samples and test reagents were thrown and allowed to reach room temperature. Then, 50 μL of the pre-diluted PACAP-38 standards and plasma samples were added to the appropriate wells of the pre-coated 96-well microplate in duplicate. For the blank wells, only the sample diluent was added to the plate. Then, 100 μL of horseradish peroxidase conjugate was added to each well, and the kit was incubated at 37 °C in the dark for 60 min. After incubation, the plate was manually washed with 200 μL of wash buffer, and the procedure was repeated four times. Next, 50 μL of chromogen solution A and 50 μL of chromogen solution B were added to each well of the plate and incubated for 15 min at 37 °C in the dark. Once the appropriate color reaction occurred, the enzyme reaction was stopped by adding 50 μL of stop solution and the optical density (OD) was read within 15 min at a wavelength of 450 nm. The SPECTROStar Nano spectrophotometer (BMG Labtech, Ortenberg, Germany) was used to measure the OD of the test wells. Since the obtained OD values were proportional to the content of PACAP-38 in the test samples, their concentrations were calculated by comparing the OD values of the sample wells with the ODs of the standard curve. All measured PACAP-38 concentrations in plasma were expressed in pg/mL.

### 4.4. Statistical Analysis

GraphPad Prism version 9.5.0 Windows program (GraphPad Software, San Diego, CA, USA, www.graphpad.com, accessed on 21 March 2022) and the MedCalc 16.8 Windows program (MedCalc Software Ltd., Ostend, Belgium, www.medcalc.org, accessed on 10 August 2016) were used for the statistical analysis. Statistical operations were performed first by performing a descriptive analysis and then by determining the distribution of samples. Outliers were identified using the robust regression and outlier removal (ROUT) method and excluded from the finer-grained statistical analysis. For comparisons between groups, the unpaired *t* test, Mann–Whitney U test, one-way ANOVA test, Kruskal–Wallis test, and mixed-effect analysis with Tukey’s, Dunn’s, or Holm–Sídák multiple comparison analysis were used, depending on the number of the compared groups, the distribution of the datasets (normal or not normal), and the types of the variables (dependent or independent). To compare two or more normally distributed groups with different homogeneity of variance, Welch or Geisser–Greenhouse correction were applied. Pearson (normal distribution) and Spearman (non-normal distribution) correlation analyses were performed to examine the correlations between parameters. The correlation coefficient (*r*) was used to determine the strength and direction of the linear relationship between variables. The receiver operating characteristic (ROC) curve analysis was used to assess the overall diagnostic performance of the PACAP-38 ELISA and to determine the cut-off value, specificity, and sensitivity of the assay. In all cases, *p* < 0.05 was considered statistically significant.

## 5. Conclusions

In conclusion, lower PACAP levels detected in MM and the further significant decrease in peptide levels in active disease support our hypothesis that PACAP may play an important role in MM. We suggest that the differences seen in the patients described in our study are due to a complex microenvironmental effect induced by plasma cells that results in reduced PACAP levels. This mechanism could lead to an additional survival advantage for plasma cells that abrogates the antitumor effect of PACAP and further worsens the condition of patients with progression. In addition, the significant prognostic role of the peptide is supported by the fact that with an increasing depth of therapeutic response, the PACAP levels of MM patients will rise until reaching the plasma values of healthy subjects. Our hypothesis is further supported by the finding that PACAP is also associated with the patients’ age, survival, plasma cell percentage in bone marrow, B2M, BJ protein, LDH, and ISS. Based on our results, the diagnostic value of PACAP in this disease is outstanding. We conclude that PACAP (along with other markers) may be a promising biomarker in the future to aid in the diagnosis of MM, assess prognosis, and potentially monitor the efficacy of clinical therapy.

## Figures and Tables

**Figure 1 ijms-24-10801-f001:**
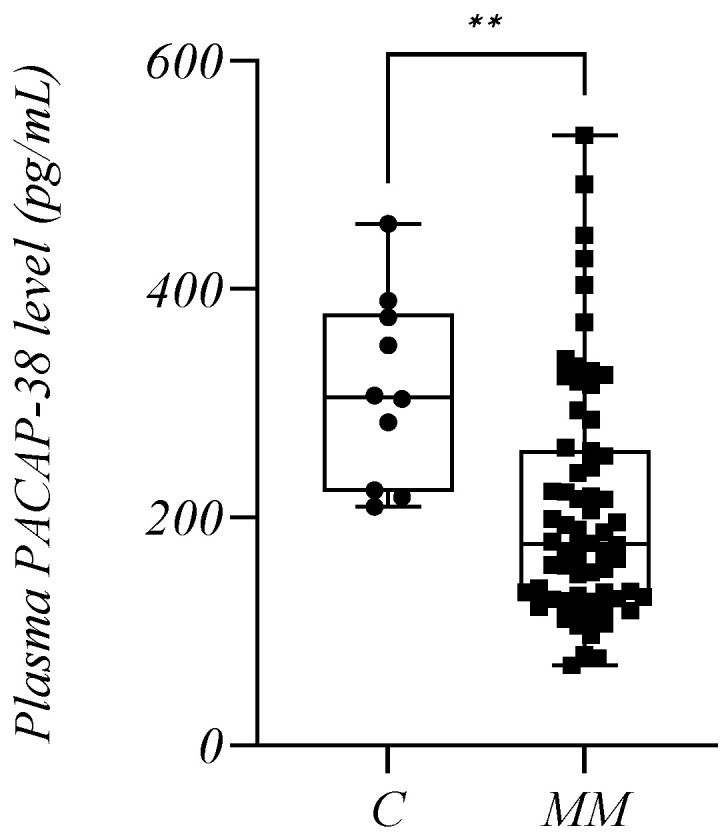
The plasma PACAP-38 levels in the multiple myeloma (MM) patients and healthy controls (C). The box plot diagram represents the interquartile range and median values. Whiskers indicate the most extreme observations. The individual values are presented with black dots (control group, *n* = 10) and squares (MM patients, *n* = 66). The Mann–Whitney U test was used for statistical analysis. ** *p* ≤ 0.01.

**Figure 2 ijms-24-10801-f002:**
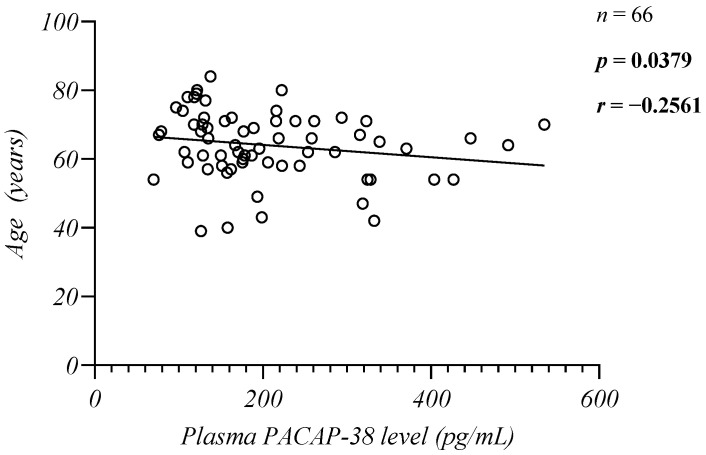
Correlation between the endogenous PACAP-38 levels and the patients’ age. The Spearman rank correlation test was used for the analysis.

**Figure 3 ijms-24-10801-f003:**
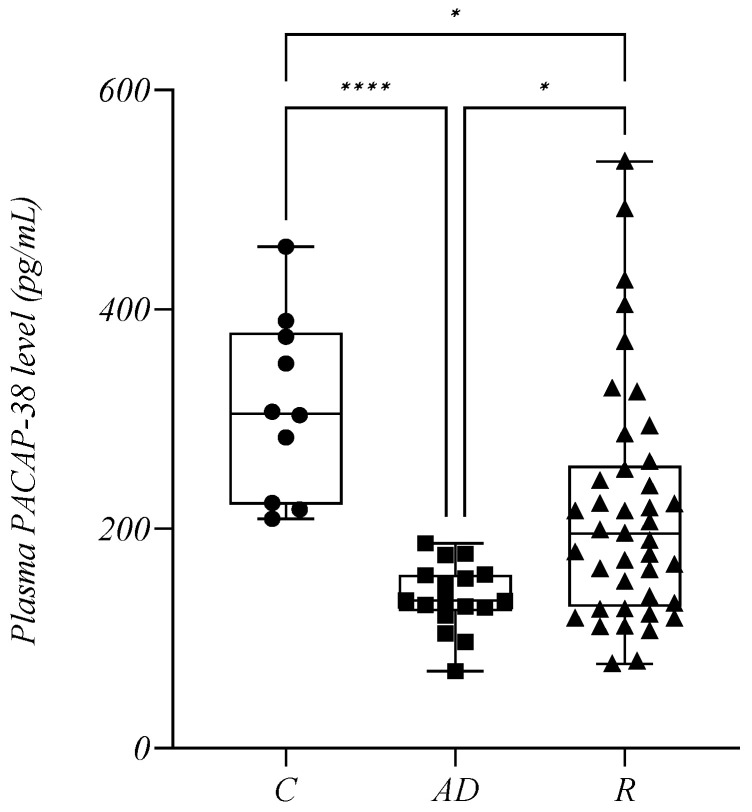
The plasma PACAP-38 levels in the control group (C), in patients with active disease (AD) and in remission (R). The box plot diagram represents the interquartile range and median values. Whiskers indicate the most extreme observations. The individual values are presented with black dots (control group, *n* = 10), squares (patients with AD, *n* = 17), and triangles (patients in R, *n* = 41). The Kruskal–Wallis with Dunn’s multiple comparison test was used for statistical analysis. * *p* ≤ 0.05, **** *p* ≤ 0.0001.

**Figure 4 ijms-24-10801-f004:**
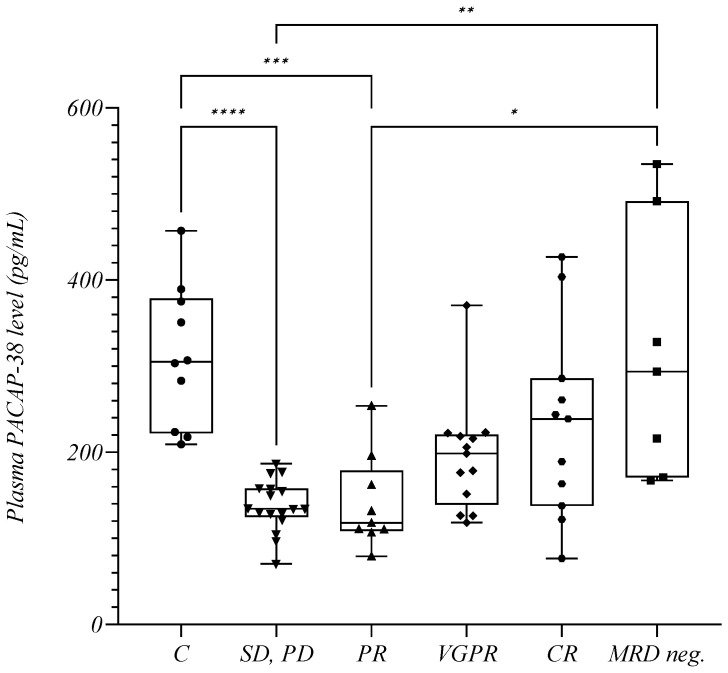
The plasma PACAP-38 levels in relation to the depth of therapeutic response. The box plot diagram represents the interquartile range and median values. Whiskers indicate the most extreme observations. The individual values are presented with black dots (control group (C), *n* = 10), downward triangles (patients with stable or progressive disease (SD, PD), *n* = 17), upward triangles (patients in partial response (PR), *n* = 10), rhombuses (patients in very good partial response (VGPR), *n* = 13), hexagons (patients in complete remission (CR), *n* = 11), and squares (patients with minimal residual disease negativity (MRD neg.), *n* = 7). For statistical analysis, the Kruskal–Wallis test with Dunn’s multiple comparison was used. * *p* ≤ 0.05, ** *p* ≤ 0.01, *** *p* ≤ 0.001, **** *p* ≤ 0.0001.

**Figure 5 ijms-24-10801-f005:**
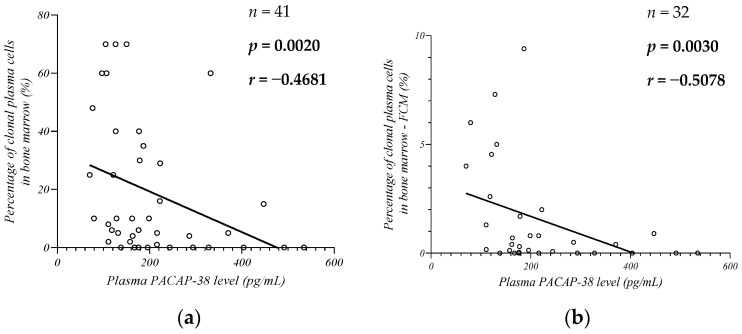
Correlation between the endogenous PACAP-38 levels and percentage of clonal plasma cells in bone marrow with histological (**a**) and flow cytometry (FCM) (**b**) examination. The Spearman rank correlation test was used for the analysis.

**Figure 6 ijms-24-10801-f006:**
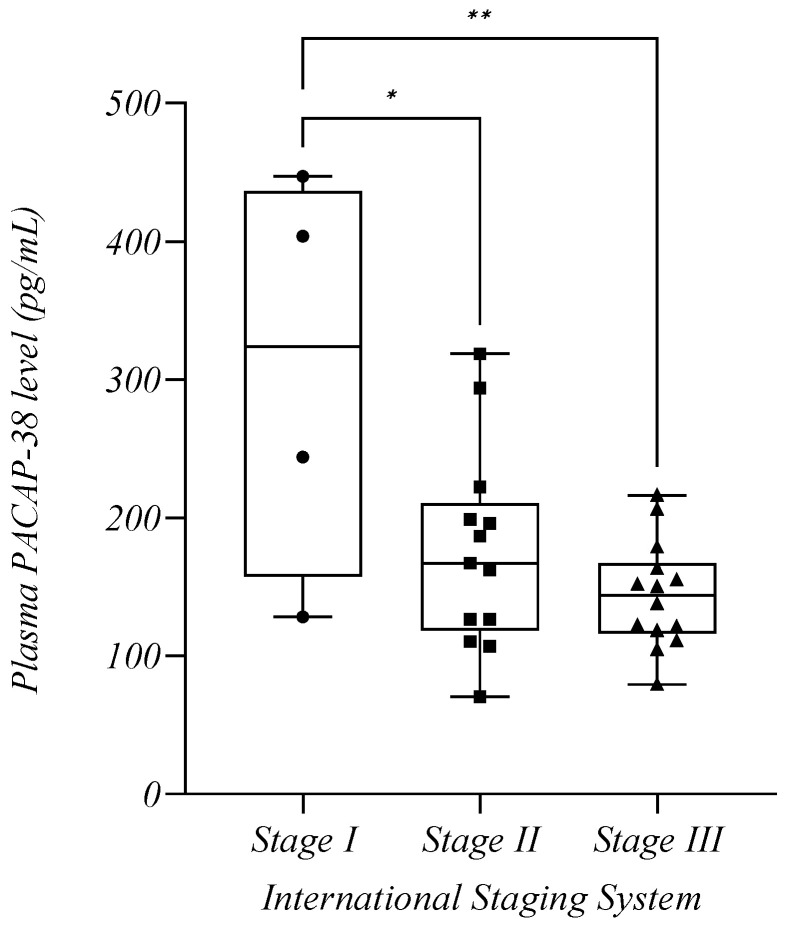
The plasma PACAP-38 levels in relation to stages of the International Staging System. The box plot diagram represents the interquartile range and median values. Whiskers indicate the most extreme observations. The individual values are presented with black dots (Stage I, *n* = 4), squares (Stage II, *n* = 13), and triangles (Stage III, *n* = 14). One-way ANOVA with Tukey’s post hoc test was used for the statistical analysis. * *p* ≤ 0.05, ** *p* ≤ 0.01.

**Figure 7 ijms-24-10801-f007:**
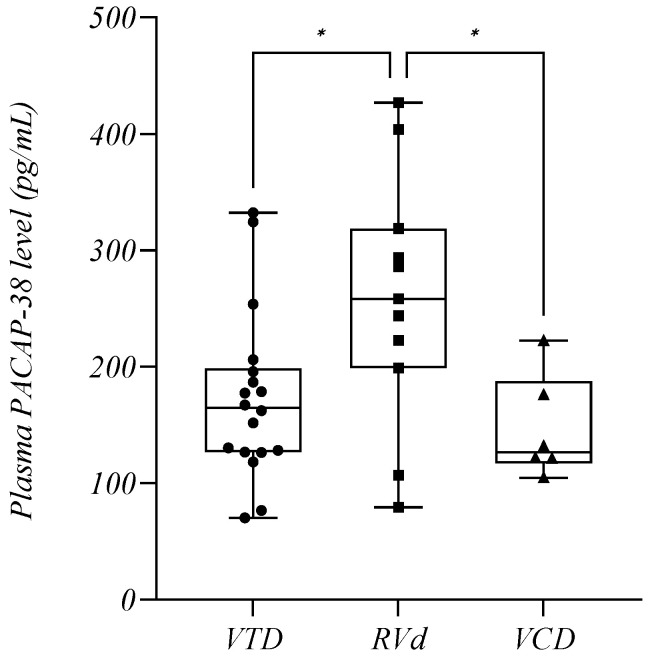
The plasma PACAP-38 levels in relation to the therapeutic combinations. The box plot diagram represents the interquartile range and median values. Whiskers indicate the most extreme observations. The individual values are presented with black dots (bortezomib–thalidomide–dexamethasone (VTD), *n* = 18), squares (lenalidomide–bortezomib–dexamethasone (RVd), *n* = 11), and triangles (bortezomib–cyclophosphamide–dexamethasone (VCD), *n* = 6). One-way ANOVA with Tukey’s post hoc test was used for the statistical analysis. * *p* ≤ 0.05.

**Figure 8 ijms-24-10801-f008:**
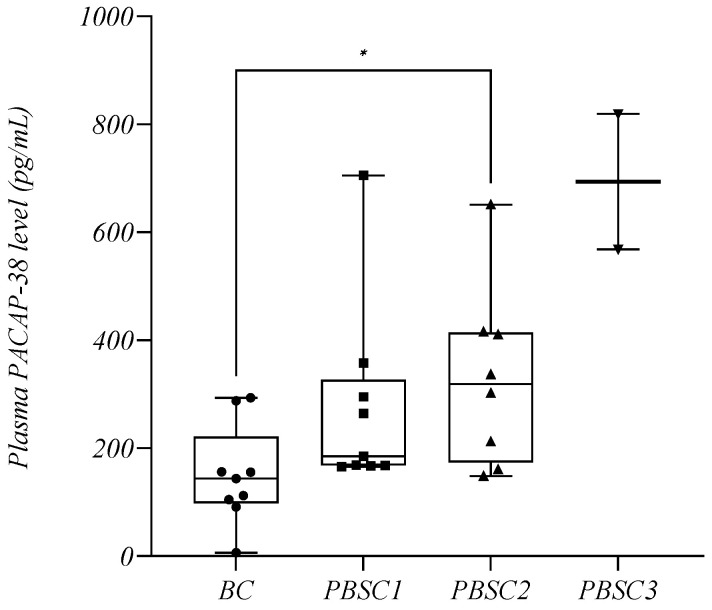
The plasma PACAP-38 levels in relation to peripheral stem cell mobilization and collection. The box plot diagram represents the interquartile range and median values. Whiskers indicate the most extreme observations. The individual values are presented with black dots (before conditioning (BC), *n* = 9), squares (peripheral blood stem cell collection (PBSC1, *n* = 9), upward triangles (PBSC2, *n* = 8), and downward triangles (PBSC3, *n* = 2). Mixed-effects analysis with Geisser–Greenhouse correction and Holm–Sídák multiple comparison test, with individual variances computed for each comparison, were used for statistical analysis. * *p* ≤ 0.05.

**Figure 9 ijms-24-10801-f009:**
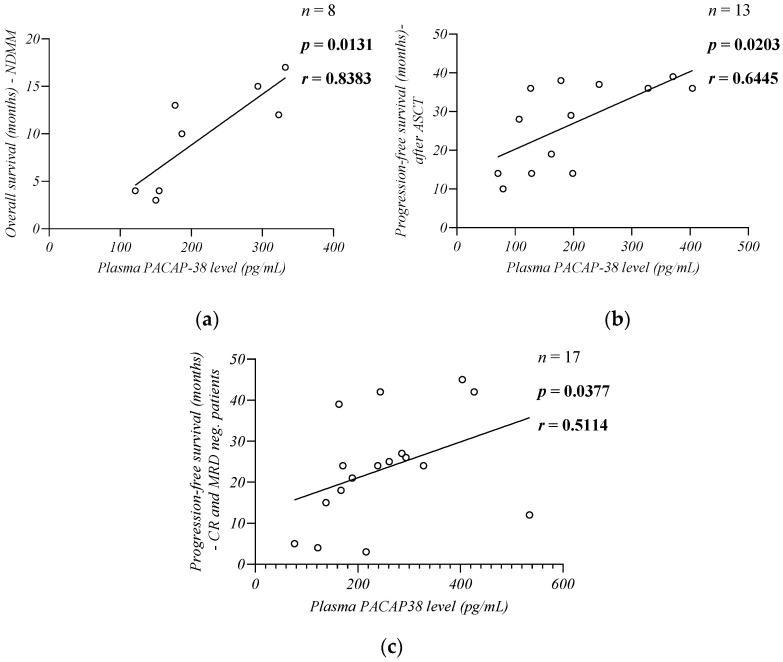
Correlation between the endogenous PACAP-38 levels and the NDMM patients’ overall survival (**a**), progression-free survival in patients who underwent autologous stem cell transplantation (ASCT) (**b**), and after reaching complete remission (CR) (**c**). The Spearman rank correlation test was used for the analysis.

**Figure 10 ijms-24-10801-f010:**
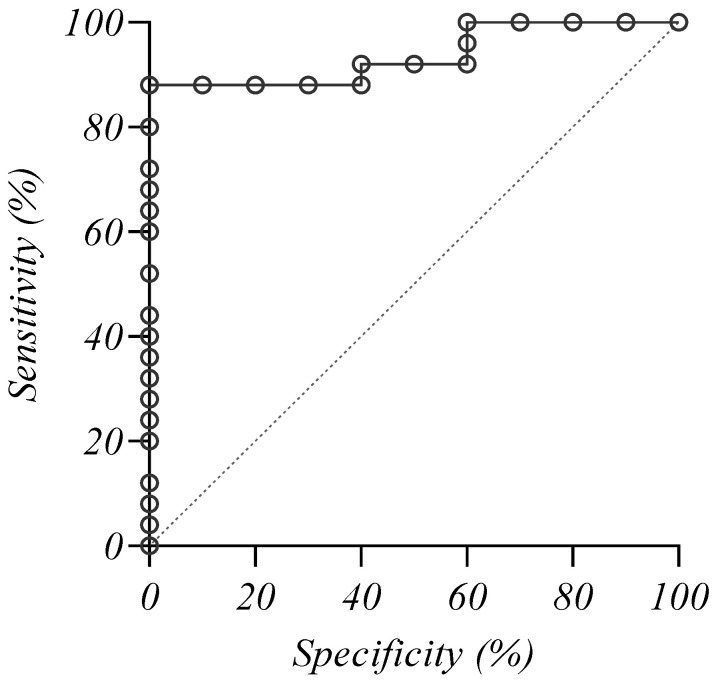
The receiver operating characteristic curve analysis of the endogenous PACAP-38 levels in myeloma patients and healthy individuals.

**Table 1 ijms-24-10801-t001:** The plasma PACAP-38 levels in relation to the comorbidities and performance status. The Mann–Whitney U and Kruskal–Wallis test with Dunn’s comparison analysis were used for the statistical analysis. ECOG: Eastern Cooperative Oncology Group, SD: standard deviation.

Tested Parameters	Patient Groups	Number of Patients (*n*)	PACAP-38 Level: Mean +/− SD (pg/mL)	Significance (*p*)
*Hypertension*	Hypertensive	28	183.3 +/− 71.41	0.3459
	Non-hypertensive	35	218.0 +/− 115.0	
*Diabetes mellitus*	Diabetic	11	265.9 +/− 139.9	0.0743
	Non-diabetic	52	185.7 +/− 80.36	
*ECOG* *performance status*	0	16	240.0 +/− 81.74	*p* _(0 vs. 1)_ = 0.7593 *p* _(0 vs. 2)_ > 0.9999 *p* _(1 vs. 2)_ = 0.3661
	1	26	174.6 +/− 83.69	
	2	14	237.1 +/− 116.0	
	3	1	338.8 +/− 0.00	

**Table 2 ijms-24-10801-t002:** The plasma PACAP-38 levels in relation to end-organ damage. The Mann–Whitney U test and Kruskal–Wallis test with Dunn’s comparison analysis were used for the statistical analysis. AL: amyloid light-chain; CKD: chronic kidney disease; EMD: extramedullary disease; PCL: plasma cell leukemia; SD: standard deviation.

Tested Parameters	Patient Groups	Number of Patients (*n*)	PACAP-38 Level: Mean +/− SD (pg/mL)	Significance (*p*)
*Bone* *lesions*	affected	35	171.1 +/− 67.64	0.0626
	non- affected	26	209.9 +/− 81.52	
*Renal* *impairment*	affected	43	207.4 +/− 97.40	0.6448
	non- affected	22	209.9 +/− 119.8	
*Acute renal* *failure*	affected	6	172.7 +/− 86.91	0.3108
	non- affected	59	206.4 +/−97.19	
*Chronic* *kidney* *disease*	normal	22	209.9 +/− 119.8	*p* _(between all subgroups)_ > 0.9999
	CKD1	4	192.5 +/− 52.53	
	CKD2	13	212.7 +/− 108.6	
	CKD3	19	219.1 +/− 105.4	
	CKD4	3	153.9 +/− 21.74	
	CKD5	4	189.5 +/− 106.7	
*Anemia*	anemic	12	194.5 +/− 93.06	0.6898
	non- anemic	52	199.8 +/− 89.86	
*EMD or PCL*	affected	18	233.9 +/− 130.5	0.3488
	non- affected	48	198.8 +/− 91.63	
*AL* *amyloidosis*	affected	4	254.3 +/− 87.36	0.1728
	non- affected	62	205.0 +/− 104.7	

**Table 3 ijms-24-10801-t003:** The plasma PACAP-38 levels in relation to stage and risk stratification. One-way ANOVA with Tukey’s post hoc test and unpaired *t* test with Welch’s correction were used for the statistical analysis. R-ISS: Revised International Staging System, SD: standard deviation.

Tested Parameters	Patient Groups	Number of Patients (*n*)	PACAP-38 Level: Mean +/− SD (pg/mL)	Significance (*p*)
*R-ISS*	Stage I	4	289.2 +/− 162.8	*p* _(Stage I vs. II)_ = 0.1547 *p* _(Stage I vs. III)_ = 0.8807 *p* _(Stage II vs. III)_ = 0.3914
	Stage II	14	183.9 +/− 74.56	
	Stage III	4	256.6 +/− 85.95	
*Cytogenetic risk*	Standard	14	233.0 +/− 103.5	0.3207
	High	13	194.1 +/− 96.13	

**Table 4 ijms-24-10801-t004:** The plasma PACAP-38 levels in relation to the laboratory parameters. The Spearman correlation analysis was used for the analysis. B2M: beta-2-microglobulin, BJ: Bence–Jones, ESR: erythrocyte sedimentation rate, LDH: lactate-dehydrogenase, k: kappa light chain, l: lambda light chain, KLR: kappa/lambda ratio, sFLC: serum free light-chains assay, STP: serum total protein, UTP: urinary total protein.

Tested Parameters	Number of Patients (*n*)	Correlation Coefficient (*r*)	Significance (*p*)
*B2M* (*all values*, mg/L)	37	−0.09627	0.5709
***B2M* (*>5.5* mg/L)**	9	**−0.7500**	**0.0255**
*LDH* (U/L)	50	−0.1472	0.3076
*STP* (g/L)	52	0.06989	0.6225
*Albumin* (g/L)	53	−0.09439	0.5014
*M-protein* (g/L)	38	−0.04738	0.7657
*sFLC: KLR—overproduction: l (<0.26)*	8	0.5952	0.1323
*sFLC: KLR—normal range (0.26–1.65)*	22	0.1249	0.5796
*sFLC: KLR—overproduction: k (>1.65)*	18	−0.3849	0.1144
*UTP* (g/L)	23	−0.4191	0.0522
***BJ Protein* (g/L)**	19	**−0.5359**	**0.0180**
*ESR* (mm/h)	25	−0.3531	0.0833

**Table 5 ijms-24-10801-t005:** The plasma PACAP-38 levels in relation to the laboratory parameters. The Mann–Whitney U test was used for the statistical analysis. LDH: lactate-dehydrogenase, SD: standard deviation, UTP: urinary total protein.

Tested Parameters	Patient Groups	Number of Patients (*n*)	PACAP-38 Level: Mean +/− SD (pg/mL)	Significance (*p*)
** *LDH* **	Below 225 U/L	7	249.4 +/− 64.19	**0.0339**
	Above 225 U/L	43	187.2 +/− 84.92	
** *UTP* **	Below 0.2 g/L	8	344.7 +/− 185.9	**0.0282**
	Above 0.2 g/L	15	193.1 +/− 78.89	

**Table 6 ijms-24-10801-t006:** The plasma PACAP-38 levels in relation to the therapeutic agents. The Mann–Whitney U test was used for the statistical analysis. IMiDs: immunomodulatory agents, SD: standard deviation.

Tested Parameters	Patient Groups	Number of Patients (*n*)	PACAP-38 Level: Mean +/− SD (pg/mL)	Significance (*p*)
*IMiDs*	Treated	37	199.0 +/− 87.54	0.0558
	Untreated	21	154.3 +/− 40.80	
** *Lenalidomide* **	Treated	18	228.6 +/− 95.63	**0.0062**
	Untreated	40	161.7 +/− 55.43	
*Thalidomide*	Treated	14	164.9 +/− 38.48	0.3083
	Untreated	44	198.6 +/− 82.59	
*Proteasome inhibitor*	Treated	44	185.8 +/− 74.11	0.4443
	Untreated	14	198.7 +/− 66.63	
** *Alkylating agents* **	Treated	10	141.3 +/− 38.16	**0.0200**
	Untreated	48	192.8 +/− 71.95	
*Steroid*	Treated	45	181.0 +/− 66.77	0.1582
	Untreated	13	225.6 +/− 104.8	
*Daratumumab*	Treated	4	289.6 +/− 150.7	0.0773
	Untreated	54	174.3 +/− 64.45	

**Table 7 ijms-24-10801-t007:** Table of disease characteristics of the patients studied. NDMM: newly diagnosed multiple myeloma; AD: active disease; SD: stable disease; PD: progressive disease; PR: partial response; VGPR: very good partial response; CR: complete remission; MRD neg.: minimal residual disease negativity; VTD: bortezomib–thalidomide–dexamethasone; RVd: lenalidomide–bortezomib–dexamethasone; VCD: bortezomib–cyclophosphamide–dexamethasone.

Characteristics of the Cohort					
**All patients (*n* = 66)**					
*NDMM (naive after diagnosis)* 8/66—12.1%			*Non-NDMM (treated)* 58/66—87.9%		
**Active or non-active disease**					
*AD*—*without NDMM* 17/58—29.3%			*Non-AD* 41/58—70.7%		
**Disease status in relation to the therapeutic response**					
*SD* 3/58—5.2%	*PD* 14/58—24.1%	*PR* 10/58—17.2%	*VGPR* 13/58—22.4%	*CR* 11/58—19%	*MRD-neg.* 7/58—12.1%
**ISS**					
*Stage I* All: 8/57—14% Analyzed: 4/31		*Stage II* All: 24/57—42.1% Analyzed: 13/31		*Stage III* All: 25/57—43.9% Analyzed: 14/31	
**R-ISS**					
*Stage I* *All: 5/44—11.4* Analyzed: 4/31		*Stage II* *All: 24/44—54.5%* Analyzed: 13/31		*Stage III* *All: 15/44—34.1%* Analyzed: 14/31	
**Cytogenetic risk**					
*Standard* All: 22/48—45.8% Analyzed: 14/27—51.9%			*high* All: 26/48—50% Analyzed: 13/27—44.4%		
**Treatments**					
*Off treatment* 0/58—0%			*On active treatment* 58/58—100%		
**Induction therapy**					
*VTD* 37/66—56.1%		*RVd* 3/66—4.5%	*VCD* 11/66—16.7%	*Other* 15/66—22.7%	
**Salvage or second line therapy**					
*VTD* 4/34—11.8%		*RVd* 8/34—23.5%	*VCD* 6/34—17.6%	*Other* 16/34—47.1%	
**Autologous stem cell transplantation**					
*Transplanted* 17/66—25.8% single: 17/17—100% tandem: 0/17—0%			*Non-transplanted* 49/66—74.2% candidate for transplant: 25/49—51%		

## Data Availability

The data presented in this study are available in the article.

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
