# Peer review of "Diagnostic and Prognostic Value of PACAP in Multiple Myeloma"

_ijms, 2023, doi:10.3390/ijms241310801_

Round 1

Reviewer 2 Report

The manuscript is very interesting, rich in data and convincing in its conclusions. I found no weak points and believe it is publishable in this form.
My only request concerns the histological analyses that are mentioned in line 214 and that also in the materials and methods are not described in any way (line 609): the authors should better specify what analysis it is

Reviewer 3 Report

Numerous articles have been published on PACAP-38 in plasma and this article concerns multiple myeloma (MM).  By ELISA, PACAP-38 levels decreased approximately 30% in 66 MM patients relative to 10 controls.  Previous low levels of PACAP-38 were found in colon carcinoma, non-small cell lung cancer, renal tumors, thyroid carcinoma, pituitary adenomas and pancreatic adenocarcinomas.  PACAP-38 was not altered as a function of sex or age, but did correlate with progression free survival after autologous stem cell transplantation.  The plasma PACAP-38 levels were lower in MM patients with active disease relative to those patients in remission.  The manuscript is well written with good statistical analysis, however, the number of patients needs to be increased.  Major revision is required.

1.  Low numbers of patients (<10) are plotted in parts of Fig. 5, 7 and 8 (Mislabeled Fig. 9).  These figures need to be redone with more patients.

2. PACAP-38 levels correlate with prognosis of MM, however, more patients are needed to evaluate diagnosis.  The sensitivity and specificity need to be determined.  Because PACAP-38 levels are lower in other types of cancer, there may be a problem with false positives.

Round 2

Reviewer 1 Report

Dear Authors,

Please see attachment. Thank you

Author Response

Dear Reviewer,

Kind regards,

The authors
